# Motion4D: Learning 3D-Consistent Motion and Semantics for 4D Scene Understanding

**Haoran Zhou**
Department of Computer Science
National University of Singapore
haoran.zhou@u.nus.edu

**Gim Hee Lee**
Department of Computer Science
National University of Singapore
gimhee.lee@nus.edu.sg

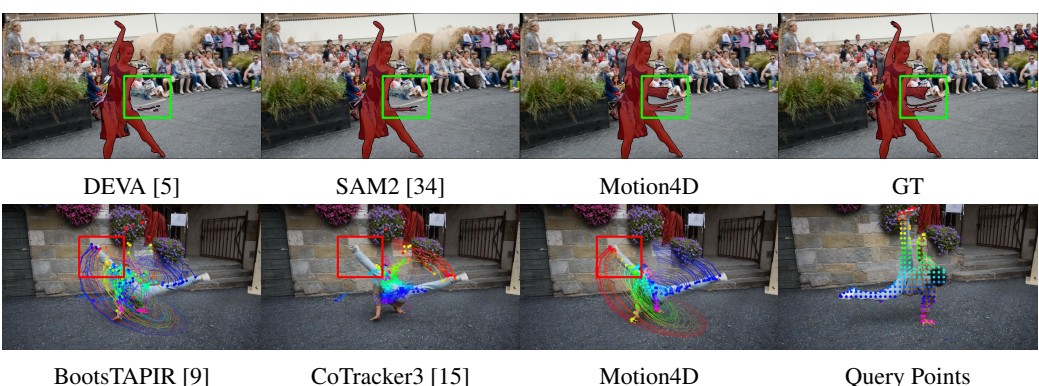

| DEVA [5] | SAM2 [34] | Motion4D | GT |

| BootsTAPIR [9] | CoTracker3 [15] | Motion4D | Query Points |

Figure 1: **Visualization of segmentation and tracking results.** We compare Motion4D with state-of-the-art 2D foundation models, highlighting their lack of 3D consistency. As shown, existing 2D approaches often suffer from temporal flickering (green box) or spatial misalignment (red box).

## Abstract

Recent advancements in foundation models for 2D vision have substantially improved the analysis of dynamic scenes from monocular videos. However, despite their strong generalization capabilities, these models often lack 3D consistency, a fundamental requirement for understanding scene geometry and motion, thereby causing severe spatial misalignment and temporal flickering in complex 3D environments. In this paper, we present Motion4D, a novel framework that addresses these challenges by integrating 2D priors from foundation models into a unified 4D Gaussian Splatting representation. Our method features a two-part iterative optimization framework: 1) Sequential optimization, which updates motion and semantic fields in consecutive stages to maintain local consistency, and 2) Global optimization, which jointly refines all attributes for long-term coherence. To enhance motion accuracy, we introduce a 3D confidence map that dynamically adjusts the motion priors, and an adaptive resampling process that inserts new Gaussians into under-represented regions based on per-pixel RGB and semantic errors. Furthermore, we enhance semantic coherence through an iterative refinement process that resolves semantic inconsistencies by alternately optimizing the semantic fields and updating prompts of SAM2. Extensive evaluations demonstrate that our Motion4D significantly outperforms both 2D foundation models and existing 3D-based approaches across diverse scene understanding tasks, including point-based tracking, video object segmentation, and novel view synthesis. Our code is available at https://hrzhou2.github.io/motion4d-web/.

39th Conference on Neural Information Processing Systems (NeurIPS 2025).

# 1  Introduction

The joint modeling of geometry, semantics and temporal dynamics of 4D scenes is a fundamental task in computer vision with broad applications in robotics, autonomous driving, augmented reality, *etc*. A key aspect of this problem is the estimation of motion-related priors, such as semantic masks, 2D point tracks, and depth maps, which serve as essential cues for reconstructing and interpreting complex scenes. However, despite significant progress, ensuring consistency and accuracy in the predictions remains challenging due to occlusions, view variations, and motion ambiguity.

Recently, the emergence of the Segment Anything Model (SAM) [18] has revolutionized 2D visual understanding and established itself as a foundational model for image segmentation. Inspired by its success, many related vision models have been developed. For example, Track Any Point [7, 8] for point-based object tracking and Depth Anything [47, 48] for monocular depth estimation. Although these models have achieved impressive performance by using large-scale datasets and extensive pre-training, they remain inherently limited in maintaining 3D coherence. In practice, even the state-of-the-art SAM2 [34] model still suffers from significant inconsistencies across frames, including spatial misalignment, temporal flickering, and boundary artifacts (*cf.* Figure 1). Such limitations arise from their design, which relies on frame-wise processing and lacks explicit 3D reasoning.

One promising solution is to *lift* 2D vision models for 3D understanding by leveraging explicit 3D representations such as Neural Radiance Fields (NeRFs) [26] or 3D Gaussian Splatting (3DGS) [17]. Unfortunately, most existing methods [52, 1, 37, 51, 2] are specifically designed for static scenes, where they improve consistency across the scene by incorporating multi-view segmentation results into an optimized 3D representation. Applying 2D models to dynamic environments introduces new challenges, including motion complexity, occlusions, and spatio-temporal alignment. These challenges, which are already problematic in 2D, remain difficult to resolve through direct integration with 3D representations. Another limitation of recent works [40, 14, 21] is that they treat 2D priors separately from dynamic 3D representations: they either learn the feature fields independently of the 3D model or decouple semantic understanding from motion estimation. As a result, the limited modeling of appearance and motion ultimately prevents the estimation of coherent predictions.

In this paper, we address the challenges of inconsistent 2D predictions and weak 3D integration by developing a unified dynamic representation that models motion and semantics from casual monocular videos. We introduce Motion4D, a method that refines multiple scene characteristics, including semantic masks, 2D point tracks, and depth estimation, while jointly optimizing a 3D Gaussian Splatting model augmented with semantic and motion fields. Motion4D consists of a two-part iterative optimization framework: 1) **Sequential optimization** that updates the motion and semantic fields in two consecutive stages within short temporal windows to maintain local consistency. 2) **Global optimization** that performs a joint optimization of all attributes throughout the sequence to ensure coherence. In the first stage of sequential optimization, we propose *iterative motion refinement* to enforce 3D motion consistency by correcting accumulated errors across sequences using a tracking loss with learned 3D confidence maps. We further introduce an *adaptive resampling module* to improve motion consistency by inserting new Gaussians into underrepresented regions identified through per-pixel RGB and semantic errors. In the second stage, we propose an *iterative semantic refinement* process to iteratively update 2D semantic priors by aligning rendered 3D masks with 2D predictions through bounding boxes and prompt points.

We introduce DyCheck-VOS, a new benchmark for evaluating video object segmentation in realistic and dynamic scenes with camera and object motion. Extensive evaluations on benchmark datasets show that our Motion4D demonstrates superior performance across various dynamic scene understanding tasks. Our method significantly outperforms both 2D foundation models and 3D representation-based methods in video object segmentation, point-based tracking, and novel view synthesis. Figure 1 shows an example of our qualitative result compared to existing approaches. Our contributions are summarized as:

- We propose Motion4D, a model that integrates 2D priors from foundation models into a dynamic 3D Gaussian Splatting representation to achieve consistent motion and semantic modeling from monocular videos.

- We design a two-part iterative optimization framework comprising sequential optimization, which updates motion and semantic fields in consecutive stages to maintain local consistency, and global optimization, which jointly refines all attributes to ensure long-term coherence.

- We introduce iterative motion refinement using 3D confidence maps and adaptive resampling to enhance dynamic scene reconstruction, and semantic refinement to correct 2D semantic inconsistencies through iterative updates with SAM2.

- Our Motion4D significantly outperforms both 2D foundation models and existing 3D methods in tasks, including video object segmentation, point-based tracking, and novel view synthesis.

## 2   Related Work

**2D vision models for scene understanding.** As a fundamental aspect of scene understanding, semantic segmentation remains an essential yet challenging task in computer vision. Recently, the Segment Anything Model (SAM) [18] was introduced as a promptable segmentation network trained on billions of masks, enabling strong zero-shot segmentation performance on new images. Its successor, SAM2 [34], extends this paradigm to video by using a streaming memory mechanism, and builds the largest video segmentation dataset (SA-V) to date. SAM2 serves as a foundation model for segmentation across images and videos, achieving state-of-the-art results in video object segmentation [5, 4, 30, 28] and offering broad applications across various domains.

Inspired by its success, many related vision models have been developed, further exploring foundation models for vision understanding. Among them, Track Any Point (TAP) [7] introduces the task of point-level tracking for dynamic scenarios, standing out as a key task in scene understanding. Building on prior work in 2D optical flow [39], it further extends the definition to capture dense and long-term relationships for tracking arbitrary points in videos. Recently, there has been a rising interest in this problem, with several works demonstrating impressive long-term 2D tracking results on challenging, in-the-wild videos [8, 19, 20]. Another important topic is the Monocular Depth Estimation (MDE) from images and videos [33, 3, 44]. This line of work aims to predict depth information for any images under any circumstances, demonstrating strong generalization abilities suitable for various downstream scenarios. Recently, Depth Anything [47] has achieved significant progress by greatly scaling up its dataset with large-scale unlabeled data, leading to significant improvements in depth estimation. Building on this progress, several works have been proposed [48, 47], with further improvement in video depth estimation and data acquisition. While these 2D foundation models demonstrate superior generalization abilities, they are still intrinsically limited in producing 3D-consistent estimations essential for scene understanding. Our work, therefore, builds on the predictions of 2D vision models and seeks to construct a joint and spatio-temporally coherent representation of multiple motion priors.

**3D representation models.** In the field of 3D vision, recent advances in 3D representation models, such as Neural Radiance Fields (Nerf) [26] and 3D Gaussian Splatting (3DGS) [17], have achieved impressive results in novel view synthesis. Leveraging the expressive representation of 3DGS, research in scene reconstruction and view synthesis has expanded to dynamic scenes [43, 24, 49, 31, 10, 27, 11], allowing for the modeling of complex object geometry and motion. On the other hand, the use of 3D models has extended beyond view synthesis to broader tasks, such as generation [13, 23], scene understanding [46, 25], and language-related applications [32]. Integrating 2D vision models with 3D representations for scene understanding is also a common strategy, typically for segmentation tasks with multi-view inputs in static scenes [51, 2, 52, 41, 37, 1]. However, when dealing with dynamic scenes involving complex motion, fewer relevant studies have been proposed. These methods either leverage an existing dynamic 3D representation with an additional semantic field [14, 21], or decouple the modeling of semantics and motion [40]. This makes them ineffective in handling complex motion in casual videos, limiting their ability to maintain 3D consistency. Compared to previous works, our method employs an effective iterative optimization strategy to jointly model semantics and motion, ensuring 3D consistency in dynamic scenes.

## 3   Our Method: Motion4D

**Problem Definition.** The input to our Motion4D is a video sequence of $T$ posed RGB images $\{I_t\}$, and a set of priors generated by 2D pre-trained models: 1) The object masks $\mathbf{M}_t$; 2) 2D point tracks $\mathbf{U}_{t \to t'}$ from $t$ to target frame at $t'$; 3) Monocular depth $\mathbf{D}_t$, where $t$ denotes the timestep. Our goal is to estimate spatio-temporally consistent predictions of semantics $\hat{\mathbf{M}}_t$ and motion

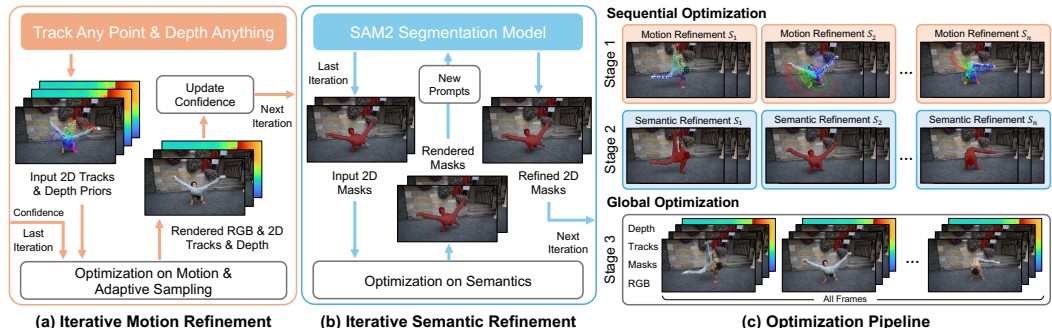

Figure 2: **Overview of Motion4D.** Our Motion4D introduces an iterative refinement framework, consisting of (c) sequential optimization and global optimization stages. We develop (a) an iterative motion refinement module that uses 3D confidence maps and adaptive resampling to improve motion accuracy, and (b) an iterative semantic refinement module to refine the semantic field.

components $\{\hat{\mathbf{U}}_{t \to t'}, \hat{\mathbf{D}}_t\}$ while simultaneously reconstructing a coherent scene representation that models appearance and dynamics of the scene.

**Overview.** Figure 2 shows an overview of our Motion4D pipeline. We first define the 3DGS with motion and semantic fields as our 4D scene understanding representation in Section 3.1, followed by the introduction of our Motion4D. Conceptually, our method aims to iteratively refine both the 2D priors and 3D scene representation throughout the optimization process of dynamic scene reconstruction. For the motion field, we update the 2D tracking priors by learning a confidence weight to control the supervision and introduce an adaptive resampling strategy to further enhance the modeling of motion (*cf.* Section 3.2). For the semantic field, we leverage the promptable SAM2 model [34] as a component in our training pipeline. After each iteration, we render semantic views from the GS model and feed them as additional prompts to the SAM2 model, which directly refines the 2D semantic priors (*cf.* Section 3.3). Furthermore, we adopt a sequential optimization strategy for motion and semantic fields to effectively mitigate error accumulations over time (*cf.* Section 3.4). Finally, we further improve reconstruction accuracy via a global optimization stage to ensure consistency and coherence across all fields (*cf.* Section 3.4). By combining the strong temporal consistency of explicit 3D representations and the rich semantic priors provided by 2D networks, our Motion4D achieves improved motion and semantic coherence across space and time.

## 3.1 4D Scene Understanding Representation

We represent the scene using 3DGS [17] and further extend it with motion and semantic fields to model dynamic 3D environments. Traditionally, 3DGS defines a set of $N$ static Gaussians in the canonical frame as $g_i^0 = \{\mu_i^0, R_i^0, s_i, o_i, c_i\}$, where $i = 1, ..., N$. $\mu_i^0 \in \mathbb{R}^3$ and $R_i^0 \in \mathbb{SO}(3)$ denote the 3D position and orientation in the canonical frame, and $s_i \in \mathbb{R}^3$ is the scale, $o_i \in \mathbb{R}$ is the opacity, and $c_i \in \mathbb{R}^3$ is the color. During rendering, given a pixel $p$ in view $\mathbf{I}$ with extrinsic matrix $\mathbf{E}$ and intrinsic matrix $\mathbf{K}$, its color $\mathbf{I}(p)$ can be computed by blending intersected 3D Gaussians:

$$\mathbf{I}(p) = \sum_{i \in H(p)} c_i \alpha_i \prod_{j=1}^{i-1} (1 - \alpha_j), \tag{1}$$

where $H(p)$ is the set of Gaussians intersecting at pixel $p$, and $\alpha_i = o_i \cdot \exp(-\frac{1}{2}(p - \mu^{2d})^\top \Sigma^{2d}(p - \mu^{2d}))$. $\mu^{2d}$ and $\Sigma^{2d}$ are the projected 2D Gaussian mean and covariance, respectively.

To model the motion of dynamic objects, we define a deformation field (motion field) which adjusts positions and orientations at each frame via rigid transformations [42]. Specifically, the pose parameters $(\mu_i^t, R_i^t)$ are rigidly transformed from the canonical frame $t_0$ via $\mathbf{T}_i^{0 \to t} = [R_i^{0 \to t} \mathbf{t}_i^{0 \to t}] \in \mathbb{SE}(3)$ for a dynamic 3D Gaussian at time $t$, *i.e.*:

$$\mu_i^t = R_i^{0 \to t} \mu_i^0 + \mathbf{t}_i^{0 \to t}, \qquad R_i^t = R_i^{0 \to t} R_i^0. \tag{2}$$

Instead of defining per-frame motion parameters for all Gaussians, we use a set of global motion bases $\{\hat{\mathbf{T}}_b^{0 \to t}\}_{b=1}^B$ and assign coefficients $w_i^b$ to each Gaussian. As a result, the per-frame transformation is obtained by a weighted combination as $\mathbf{T}_i^{0 \to t} = \sum_{b=0}^B w_i^b \hat{\mathbf{T}}_b^{0 \to t}$.

Due to the explicit representation of 3DGS, we follow [51, 2, 21] to directly embed a semantic field onto each Gaussian for scene semantics. This is analogous to the role of the color field during rendering. As a result, we can render the semantic view in a similar way as Equation 1 by:

$$\mathbf{I}^{\text{sem}}(p) = \sum_{i \in H(p)} f_i^{\text{sem}} \alpha_i \prod_{j=1}^{i-1} (1 - \alpha_j). \tag{3}$$

This gives the per-pixel semantic features, which can be converted to object masks $\hat{\mathbf{M}}_t$ at time $t$.

## 3.2 Iterative Motion Refinement

Figure 2(a) illustrates the motion field refinement process, where we focus on improving the motion (deformation field) of dynamic Gaussian Splatting that directly impacts the accuracy of 3D point tracking estimation. The initial 3D point tracks are formulated by the 2D point tracks $\mathbf{U}_{t \to t'}$ and monocular depth $\mathbf{D}_t$. Following [42], we then compute the pixel-wise 3D motion trajectory of the query frame $t$ by rendering the 3D positions of Gaussian points at the target time $t'$:

$$\mathbf{X}_{t \to t'}(p) = \sum_{i \in H(p)} \mu_i^{t'} \alpha_i \prod_{j=1}^{i-1} (1 - \alpha_j), \tag{4}$$

where $\mu_i^{t'}$ is position of the Gaussian at the target frame. $\mathbf{X}_{t \to t'}(p)$ therefore estimates the 3D position of the pixel at time $t'$, which can be projected into 2D tracks $\hat{\mathbf{U}}_{t \to t'}(p)$ and depth $\hat{\mathbf{D}}_{t \to t'}(p)$. We now consider supervising these estimates using the corresponding 2D priors.

Unlike the refinement of semantics (*cf.* Section 3.3), 2D tracking networks do not support prompt-based interactions such as those in SAM2, *i.e.* we cannot directly update 2D track and depth priors to ensure consistency. Instead, we seek to control the supervision loss by assigning a pixel-wise confidence weight $w(p)$ to reduce the influence of erroneous estimations in the input priors:

$$L_{\text{track}} = \frac{1}{|I_t|} \sum_{p \in I_t} w(p) \|\hat{\mathbf{U}}_{t \to t'}(p) - \mathbf{U}_{t \to t'}(p)\|, \quad L_{\text{depth}} = \frac{1}{|I_t|} \sum_{p \in I_t} w(p) \|\hat{\mathbf{D}}_{t \to t'}(p) - \mathbf{D}_t(p')\|. \tag{5}$$

where $p' = \mathbf{X}_{t \to t'}(p)$ is the position at $t'$. The confidence weight $w(p)$ estimates the probability that the input priors are inconsistent with the ground truth. We then add a new uncertainty field $u_i \in \mathbb{R}$ to each Gaussian and obtain the weight by rendering the 3D uncertainty logits into 2D:

$$w(p) = \sum_{i \in H(p)} u_i \alpha_i \prod_{j=1}^{i-1} (1 - \alpha_j). \tag{6}$$

We define the weight by evaluating the self-consistency of the pixel across time in terms of the color and semantic estimations:

$$L_w = \text{BCE}(\hat{w}(p), w(p)), \quad \text{where } \hat{w}(p) = \begin{cases} 1 & \text{if } \|\mathbf{I}_t(p), \mathbf{I}_{t'}(p')\| < \delta \text{ and } \|\mathbf{M}_t(p), \mathbf{M}_{t'}(p')\| < \delta', \\ 0 & \text{otherwise.} \end{cases} \tag{7}$$

where $\text{BCE}(\cdot)$ is the binary cross entropy, and $\delta$ and $\delta'$ are the distance thresholds.

**Adaptive Resampling.** We introduce a new 3DGS densification strategy to further enhance motion modeling through an adaptive resampling process. As discussed in prior works [35, 38], the Adaptive Density Control of 3DGS [17] relies heavily on gradient magnitude. This makes it sensitive to the choice of loss functions and often fails to identify the underfitting regions. Consequently, we extend the idea of error-based densification from static scenes [35] to dynamic environments and develop a sampling strategy to insert new Gaussians into the under-represented regions of the moving objects. In each iteration, we first compute per-pixel errors $e_{\text{rgb}}(p)$ and $e_{\text{sem}}(p)$ for the RGB and semantic views, respectively. We then select regions where $e_{\text{rgb}}(p) > \theta_{\text{rgb}}$ or $e_{\text{sem}}(p) > \theta_{\text{sem}}$. Finally, we sample 2D points from these error-prone regions, project them into 3D using the rendered depth, and initialize new Gaussians based on their nearest dynamic Gaussians. By progressively refining and filling sparsely reconstructed regions, our Motion4D effectively recovers blurry or missing parts of foreground targets that are typically caused by inaccurate initial motion estimates or tracking failures.

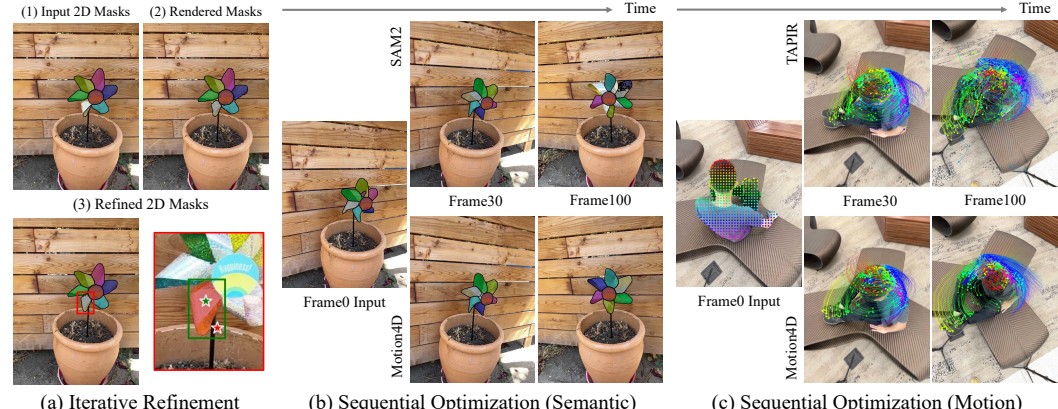

(1) Input 2D Masks  (2) Rendered Masks

(3) Refined 2D Masks

(a) Iterative Refinement

(b) Sequential Optimization (Semantic)

(c) Sequential Optimization (Motion)

Figure 3: Illustration of (a) iterative semantic refinement and sequential optimization processes for (b) the semantic field and (c) the motion field. The proposed strategies effectively update 2D input priors to achieve consistent results across space and time.

## 3.3 Iterative Semantic Refinement

Our Motion4D iteratively refines scene semantics by alternating between updating the 3D representation and semantic field, and refining the 2D semantic priors using the reconstructed 3D scene representation. Specifically, at the $s$-th iteration step, we first obtain 2D segmentation masks $\{\mathbf{M}_t^{s-1}\}_{t=1}^T$ generated by SAM2 in the previous iteration to supervise the semantic field during its optimization. We then render 3D masks $\hat{\mathbf{M}}_t^s$ and compare each rendered mask with the corresponding 2D mask $\mathbf{M}_t^{s-1}$ to identify mismatched regions that require additional prompts. For each object, $m$ prompts are generated to align the 2D predictions with the rendered 3D masks as follows: 1) an exact bounding box of the 3D mask is provided, and 2) we place positive or negative prompt points at the center of the most prominent regions, determined by the maximum value of the distance transform. Note that, we avoid using the exact 3D mask as a prompt since that SAM2 tends to strictly follow the mask input, limiting its flexibility in refining 2D semantic priors. Figure 3(a) provides a visual example of the iterative refinement on semantics. We show that rendered 3D masks offer greater consistency inherent to the 3D scene representation, while SAM2 excels at preserving fine-grained high-resolution details. To this end, our method effectively updates inconsistent 2D masks through additional prompts that resolve semantic ambiguity.

## 3.4 Optimization Pipeline

Figure 2(c) illustrates the overall optimization pipeline, which consists of the sequential and global optimization in three stages.

**Sequential Optimization.** The sequential optimization consists of two stages. In Stage 1, our Motion4D optimizes the motion field across video sequences, where each sequence is defined as a chunk of consecutive frames $\mathcal{S}_i = I_t \mid t \in [iL, (i+1)L)$. To enhance short-term tracking and dynamics, we apply the iterative motion refinement module within each sequence. We proceed to Stage 2 upon completion of the motion field optimization, which focuses on optimizing the semantic field. During this stage, we ensure a more stable semantic refinement by keeping the motion field fixed to prevent inaccurate semantic priors from affecting the Gaussian parameters.

Sequential optimization is crucial for maintaining long-term consistency in both motion and semantic estimations. As shown in Figure 3(b), 2D priors are prone to error accumulation over time since 2D networks typically rely on the short-term memory of recent predictions. We observe that the SAM2 result aligns well with objects in the initial frames but gradually loses track as the video progresses. The accumulation of these inconsistencies corrupt the semantic representation and dominate the supervision when optimizing over the entire sequence space. To address this issue, our Motion4D uses sequential optimization steps to produce consistent results over short temporal windows and progressively extending them across the entire video. To ensure consistency between sequences, the semantic refinement step explicitly updates SAM2 semantic priors to enhance temporal consistency

Table 1: Comparisons of segmentation performance on DyCheck-VOS and DAVIS. We compare the results of 2D networks (2D) and 3D methods (3D + SAM2 masks).

| Method | Representation | DyCheck-VOS | | | DAVIS 2017 val | | |
| --- | --- | --- | --- | --- | --- | --- | --- |
| | | $\mathcal{J}\&\mathcal{F}$ | $\mathcal{J}$ | $\mathcal{F}$ | $\mathcal{J}\&\mathcal{F}$ | $\mathcal{J}$ | $\mathcal{F}$ |
| XMem [4] | 2D | 83.5 | 81.0 | 86.0 | 86.2 | 82.9 | 89.5 |
| DEVA [5] | 2D | 84.5 | 81.7 | 87.4 | 87.0 | 83.6 | 90.4 |
| SAM2 [34] | 2D | 89.4 | 88.3 | 90.5 | 90.7 | 89.4 | 92.0 |
| Semantic Flow [40] | 3D + SAM2 masks | 76.9 | 74.4 | 79.3 | 72.2 | 69.3 | 75.2 |
| SADG [21] | 3D + SAM2 masks | 81.8 | 79.2 | 84.3 | 75.0 | 71.9 | 78.1 |
| Motion4D | 3D + SAM2 masks | 91.0 | 89.6 | 92.4 | 89.7 | 86.1 | 90.3 |
| Motion4D + SAM2 [34] | 3D + SAM2 masks | **91.7** | **90.4** | **93.0** | **90.8** | **89.6** | **92.0** |

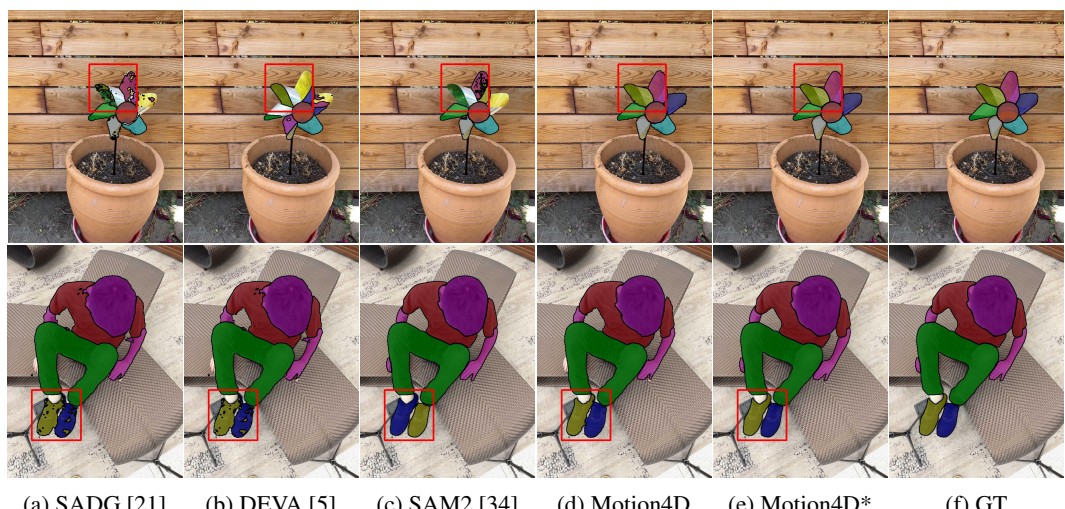

(a) SADG [21]    (b) DEVA [5]    (c) SAM2 [34]    (d) Motion4D    (e) Motion4D*    (f) GT

Figure 4: Visualization of segmentation results on DyCheck-VOS (our proposed VOS benchmark). We provide our results of both rendered masks (Motion4D) and refined SAM2 masks (Motion4D*). As shown, the 2D predictions lack 3D consistency, which leads to misaligned spatial structures.

and correct accumulated errors. The motion refinement step also corrects accumulated errors by implicitly aligning the motion trajectories through the tracking loss (Equation 5) across sequences. Concurrently, these complementary refinements ensure robust scene understanding over time.

**Global Optimization.** Finally, we perform a global optimization in Stage 3 to jointly train all fields over the full sequences of video frames. The global stage integrates information across all fields to achieve a coherent 4D scene representation. We show in Section 4.4 that it is beneficial to learn motion and semantic representations together via scene reconstruction. The overall loss is defined as $\mathcal{L} = \lambda_{\text{rgb}}L_{\text{rgb}} + \lambda_{\text{sem}}L_{\text{sem}} + \lambda_{\text{track}}L_{\text{track}} + \lambda_{\text{depth}}L_{\text{depth}} + \lambda_{w}L_{w}$, where each $\lambda$ is a hyperparameter to balance the loss terms.

## 4 Experiments

We evaluate Motion4D across diverse tasks, including video object segmentation, point-based tracking, and novel view synthesis, to demonstrate its ability to model motion and semantics in dynamic scenes.

### 4.1 Segmentation Results

**Introducing DyCheck-VOS for Video Object Segmentation.** We create a new VOS benchmark to evaluate segmentation performance in realistic and dynamic scenes by manually annotating the DyCheck dataset [12] with high-quality per-frame object masks. The DyCheck dataset was originally designed for scene reconstruction and novel view synthesis. It contains 14 sequences, each with

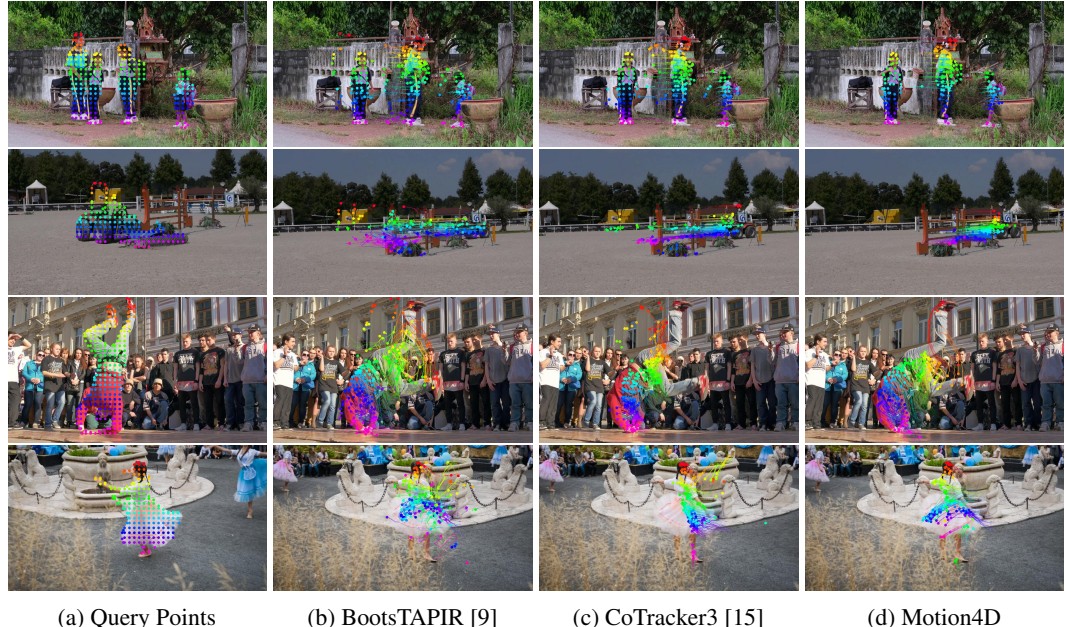

| (a) Query Points | (b) BootsTAPIR [9] | (c) CoTracker3 [15] | (d) Motion4D |

Figure 5: Visualization of 2D point tracking results. Motion4D maintains stable and accurate point trajectories even under severe occlusions and drastic object or camera motion.

200–500 frames, featuring diverse types of real-world motions. We annotate selected foreground objects of interest, with a focus on partial regions instead of full-object masks, which involve more challenging motion patterns and frequent occlusions. As shown in Figure 4, 2D networks often produce inconsistent predictions on object subparts, making DyCheck-VOS a strong benchmark for assessing semantic consistency. We follow the standard VOS settings [30, 45] and report the $\mathcal{J}\&\mathcal{F}$ which is the average of Jaccard index ($\mathcal{J}$), and boundary F1-score ($\mathcal{F}$).

We also evaluate our method on the DAVIS dataset [30]. We use the DAVIS 2017 validation set and follow the standard setting of semi-supervised video object segmentation by providing the ground-truth object masks on the first frame as input.

**Results.** Table 1 summarizes the results. We report results for both the rendered masks (Motion4D) and the refined SAM2 outputs by our method (Motion4D + SAM2) in the comparisons. For DyCheck-VOS, we show that Motion4D significantly outperforms all compared methods, including both 2D segmentation models and approaches based on 3D representations. For the compared 2D models [4, 5], we follow standard VOS settings by using the ground-truth masks of the first frame as input. For the 3D-based approaches [40, 21], we provide masks generated by SAM2 to match the input setting used in our method. Unfortunately, the full training code for SADG [21] is not publicly available, and we implement the training pipeline based on their released model code. Figure 4 shows visual comparisons of the segmentation results. We show that 2D models often struggle to maintain consistent object masks, and 3D-based methods, when given inconsistent masks as input, also fail to improve the quality, leading to degraded performance. In contrast, our method is able to produce consistent estimations by effectively combining the 2D masks and 3D representation.

We also observe strong performance on the DAVIS dataset, demonstrating the generalization of our method. As shown in Figure 1, Motion4D effectively improves the segmentation results of SAM2 by correcting temporally inconsistent regions. While the directly rendered masks from Motion4D achieve slightly lower scores, this is primarily due to limitations in reconstruction quality, which can affect fine-grained mask boundaries.

## 4.2 2D Point Tracking Results

We next evaluate our method on the task of 2D point tracking. The DAVIS dataset, introduced for 2D point tracking by TAP-Vid [7], provides sparsely annotated point trajectories across real-world

Table 2: Comparison of 2D point tracking performance on DAVIS [7] and DyCheck [12] datasets.

| Method | DAVIS 2D Tracking | | | DyCheck 2D Tracking | | |
|---|---|---|---|---|---|---|
| | AJ $\uparrow$ | $< \delta_{\text{avg}} \uparrow$ | OA $\uparrow$ | AJ $\uparrow$ | $< \delta_{\text{avg}} \uparrow$ | OA $\uparrow$ |
| TAPIR [8] | 56.2 | 70.0 | 86.5 | 27.8 | 41.5 | 67.4 |
| LocoTrack [6] | 62.9 | 75.3 | 87.2 | 26.1 | 40.5 | 66.9 |
| CoTracker [16] | 61.8 | 76.1 | 88.3 | 24.1 | 33.9 | 73.0 |
| BootsTAPIR [9] | 61.4 | 73.6 | 88.7 | 30.1 | 42.8 | 78.5 |
| CoTracker3 [15] | **64.4** | 76.9 | **91.2** | 31.0 | 44.4 | 79.9 |
| HyperNeRF [29] | 50.6 | 63.6 | 81.0 | 10.1 | 19.3 | 52.0 |
| Deformable-3D-GS [50] | 55.6 | 69.4 | 82.0 | 14.0 | 20.9 | 63.9 |
| Shape of Motion [42] | 62.3 | 76.2 | 87.1 | 34.4 | 47.0 | 86.6 |
| Motion4D | **64.4** | **77.7** | 90.4 | **37.3** | **50.4** | **87.1** |

Table 3: Comparison of 3D point tracking and novel view synthesis results on DyCheck [12] dataset.

| Method | 3D Point Tracking | | | Novel View Synthesis | | |
|---|---|---|---|---|---|---|
| | EPE $\downarrow$ | $\delta_{\text{3D}}^{.05} \uparrow$ | $\delta_{\text{3D}}^{.10} \uparrow$ | PSNR $\uparrow$ | SSIM $\uparrow$ | LPIPS $\downarrow$ |
| T-NeRF [12] | - | - | - | 15.60 | 0.55 | 0.55 |
| HyperNeRF [29] | 0.182 | 28.4 | 45.8 | 15.99 | 0.59 | 0.51 |
| DynIBaR [22] | 0.252 | 11.4 | 24.6 | 13.41 | 0.48 | 0.55 |
| Deformable-3D-GS [50] | 0.151 | 33.4 | 55.3 | 11.92 | 0.49 | 0.66 |
| CoTracker [16] + Depth Anything [47] | 0.202 | 34.3 | 57.9 | - | - | - |
| TAPIR [8] + Depth Anything [47] | 0.114 | 38.1 | 63.2 | - | - | - |
| Shape of Motion [42] | 0.082 | 43.0 | 73.3 | 16.72 | 0.63 | 0.45 |
| Motion4D | **0.072** | **46.7** | **75.9** | **17.91** | **0.69** | **0.42** |

video sequences. It serves as a benchmark for assessing a model's ability to track arbitrary points under occlusion, deformation, and complex motion. Similarly, the DyCheck [12] dataset provides annotations of 5 to 15 keypoints sampled at equally spaced time steps for each sequence. The dataset contains longer video sequences, making it well suited for evaluating long-term point tracking performance. We therefore evaluate the tracking performance in terms of both position accuracy and occlusion accuracy, reporting the Average Jaccard (AJ), average position accuracy ($< \delta_{\text{avg}}$), and Occlusion Accuracy (OA).

**Results.** Table 2 summarizes the quantitative results for point-based tracking. Motion4D achieves superior accuracy compared to both 2D tracking models and 3D-based methods across various challenging scenarios. Moreover, Figure 5 presents visual comparisons with strong 2D tracking baselines. As the ground-truth annotations in the DAVIS dataset are available only for sparsely sampled points, we visualize dense query point tracking results for qualitative comparison. Motion4D demonstrates clear advantages particularly in handling severe occlusions (first and second rows) and drastic motion (third and fourth rows), maintaining consistent and accurate trajectories throughout the sequences. In contrast, 2D tracking models tend to lose track of points under such challenging conditions, resulting in noticeable drift and fragmented tracks.

### 4.3 3D Point Tracking and Novel View Synthesis

Finally, we evaluate our method on the DyCheck [12] dataset for the tasks of 3D point tracking and novel view synthesis. Following the experimental setting of [42], we use camera poses estimated by COLMAP [36] and initial monocular depths from Depth Anything [47]. For 3D point tracking, we generate ground-truth 3D trajectories by lifting the 2D keypoint annotations into 3D using LiDAR depth, and evaluate the tracking performance using the 3D end-point-error (EPE) and the percentage of points that fall within a given threshold $\delta_{\text{3D}}^{.05} = 5\text{cm}$ and $\delta_{\text{3D}}^{.10} = 10\text{cm}$. For novel view synthesis, we assess reconstruction quality using PSNR, SSIM, and LPIPS scores.

**Results.** We report quantitative results for both tasks in Table 3. Motion4D consistently outperforms state-of-the-art methods, including both 2D- and 3D-based approaches, across the 3D tracking and novel view synthesis tasks. For 3D point tracking, our method achieves lower 3D end-point error and

Table 4: Ablation studies on the DyCheck-VOS and DyCheck dataset.

| Method | 2D Update | Sampling | Seq. Opt. | Global Opt. | $\mathcal{J}\&\mathcal{F}\uparrow$ | AJ $\uparrow$ | $< \delta_{avg}\uparrow$ | OA $\uparrow$ |
|---|---|---|---|---|---|---|---|---|
| Ours (Full) | ✓ | ✓ | ✓ | ✓ | **91.7** | **37.3** | **50.4** | **87.1** |
| w/o Iterative Refinement | | ✓ | ✓ | ✓ | 87.6 | 34.6 | 47.2 | 86.5 |
| w/o Adaptive Sampling | ✓ | | ✓ | ✓ | 88.9 | 35.1 | 47.7 | 84.2 |
| Full Initialization | ✓ | ✓ | | ✓ | 88.0 | 34.9 | 47.5 | 87.0 |
| w/o Global Optimization | ✓ | ✓ | ✓ | | 90.3 | 36.5 | 49.4 | 86.6 |

higher accuracy at both $\delta_{3D}^{.05}$ and $\delta_{3D}^{.10}$ thresholds, demonstrating more precise and temporally stable motion estimation. For novel view synthesis, Motion4D produces sharper and more geometrically consistent renderings, achieving higher PSNR and SSIM scores and lower LPIPS values, indicating improved visual fidelity.

### 4.4 Ablation Studies

We ablate key components of our method on the DyCheck dataset, with results summarized in Table 4. First, we validate the effectiveness of the iterative refinement strategy. In the ablation of "w/o Iterative Refinement", we remove the iterative update mechanism used to refine 2D priors during training, including both the update of 2D semantic masks and the confidence-based refinement of point tracks. In this setting, only the 3D scene representation is optimized, while the 2D priors remain fixed throughout training. In addition, the "w/o Adaptive Sampling" variant disables the densification step based on error-driven sampling. The results highlight the importance of both iterative refinement and adaptive sampling for achieving robust motion and segmentation performance.

Next, we validate the optimization strategies including sequential and global optimization stages. In the "Full Initialization" variant, we apply a straightforward initialization over the entire video sequence without dividing it into temporal chunks. This setting suffers from inconsistent supervision due to unreliable 2D priors, often leading to unstable training and degraded performance. In the "w/o Global Optimization" setting, we skip the final stage of joint optimization over the full sequence and instead rely solely on sequential updates. This results in local consistency but introduces slight temporal drift across chunks. As shown in Table 4, both optimization stages are critical for achieving accurate and temporally coherent motion and segmentation results.

## 5 Conclusion

We present Motion4D, a unified framework for dynamic 3D scene reconstruction that jointly models motion and semantics via 4D scene reconstruction. By leveraging iterative refinement of 2D priors and a multi-stage optimization strategy, Motion4D achieves robust and temporally consistent performance across multiple tasks, including video object segmentation, point-based tracking, and novel view synthesis. We also introduce DyCheck-VOS, an annotated benchmark for segmentation in dynamic reconstruction scenes, and demonstrate the effectiveness of our method on various tasks. While Motion4D achieves strong performance in dynamic scene understanding, it still relies heavily on the quality of the underlying 3D reconstruction. In scenes with severe occlusions, low-texture regions, or inaccurate depth estimation, the reconstruction quality can degrade, which in turn affects the accuracy of motion and semantic predictions.

## Acknowledgment

This research / project is supported by the National Research Foundation (NRF) Singapore, under its NRF-Investigatorship Programme (Award ID. NRF-NRFI09-0008), and the Tier 2 grant MOE-T2EP20124-0015 from the Singapore Ministry of Education.

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
