# OpenReview forum: "Motion4D: Learning 3D-Consistent Motion and Semantics for 4D Scene Understanding"
_NeurIPS.cc/2025/Conference — NeurIPS 2025 poster_

### Official Review · Reviewer_8CeR · 2025-06-25

**Clarity:** 3
**Significance:** 2
**Originality:** 3
**Rating:** 4
**Confidence:** 4

**Summary:**

The paper addresses a critical challenge in dynamic scene understanding: the lack of 3D consistency in 2D foundation models (e.g., temporal flickering and spatial misalignment).
To this end, this paper presents a framework that integrate 2D priors from foundation models into a unified 4D scene representation.
A two-part iterative optimization framework is designed to updates motion and semantic fields in consecutive stages and jointly refines all attributes for long-term coherence.
The paper evaluates on multiple tasks (segmentation, point tracking, novel view synthesis) and benchmarks (DyCheck-VOS, DAVIS). The results show clear improvements over existing 2D and 3D methods.

**Questions:**

See the weaknesses above.

How does the adaptive resampling strategy handle regions that are under-represented due to occlusion? Does it rely on the depth estimates, which might be inaccurate in occluded regions?

**Ethical Concerns:**

["NO or VERY MINOR ethics concerns only"]

**Final Justification:**

While I still think the improvement on DAVIS 2017 val is trivial, the rebuttal addressed most of my concerns. So I keep my rating as Borderline Accept.

**Limitations:**

Sequential-global optimization is resource-heavy and time-consuming. The 2D foundation models can be used directly without any additional iteratively optimization, even though the proposed framework improves the 2D priors with more spatial consistent results, it might be hard to support tasks requires faster inference.

**Paper Formatting Concerns:**

No major formatting issues noticed.

**Quality:**

3

**Strengths And Weaknesses:**

Strengths:

S1: The integration of 2D priors (SAM2, TAP, Depth Anything) into a dynamic 3DGS representation with iterative refinement is novel.

S2: The two-part optimization (sequential and global) is a well-motivated approach to correct 2D priors using 3D reconstruction feedback and handle error accumulation.

S3: The use of 3D confidence maps for motion refinement and adaptive resampling based on per-pixel errors is novel.

S4: A new benchmark is provided. DyCheck-VOS addresses the lack of dynamic-scene segmentation datasets.


Weaknesses:


W1: 2D Prior Dependency: Performance degrades with low-quality inputs.

W2: In Table 1, the improvement on DAVIS 2017 val is trivial, or no improvement is observed (SAM2 vs. Motion4D + SAM2).

W3: Missing ablation on optimization tasks: like only using part of the 2D tasks and foundation models, e.g., one or two for the two tasks (tracking, depth, and segmentation).

W4: The contribution of each component (3D confidence map, adaptive resampling, iterative semantic refinement) could be isolated more clearly.

---

> ### Author Rebuttal · Authors · 2025-07-31
>
> **Responses to Reviewer 8CeR**
>
> **Q1: 2D prior dependency.**
>
> **A1:** Although our method uses multiple 2D priors, our objective is to ***refine and correct these priors*** through 3D consistency and temporal coherency. As demonstrated by the results in Figure 1 and 3, our iterative optimization substantially improves the input 2D priors even when they are noisy. Our design offers a distinct advantage over both pure 2D and 3D methods. Pure 2D networks often produce inconsistent outputs, particularly in regions with severe occlusions and motion ambiguity. Furthermore, 3D baselines struggle to leverage priors as effectively as our approach. Consequently, our refinement process is essential to achieve stable and robust results despite inheriting some limitations of the 2D priors.
>
> **Q2: Improvement on DAVIS 2017 val is trivial.**
>
> **A2:** The modest gain on DAVIS 2017 val arises because the baseline (SAM2) already achieves near-saturated performance (over 90%) on this split. While diverse in content, the DAVIS dataset contains short video clips (fewer than 100 frames) with salient foreground targets and limited occlusion. In contrast, our method shows clear improvements on more challenging scenes (over 400 frames) in the DyCheck dataset (Tables 1 and 2), which require long-term tracking across extended sequences. Even on DAVIS val, our method enhances temporal consistency across frames (Figure 1), which is not fully captured by raw performance scores.
>
> **Q3: Missing ablation on optimization tasks.**
>
> **A3:** Tracking and depth cannot be treated as independent tasks as each track corresponds to a 3D trajectory. Since 3D tracks are not directly observable, we supervise the motion field using both 2D tracking and depth cues. Ablating either signal leads to a significant performance drop: using only tracking reduces $<\delta_{avg}$ on DyCheck from 50.4 to 41.6, and using only depth yields 25.9. This highlights that both signals are essential and complementary. Similarly, training solely on segmentation is not viable since the semantic field relies on the underlying 3D geometry. Without the motion field (tracking and depth), the model fails to converge and produces unusable results. We will include these additional ablations in the final paper to clarify the necessity of the full design.
>
> **Q4: The contribution of each component.**
>
> **A4:** The effectiveness of our adaptive resampling module has been shown in the ablation study in Table 3 (“w/o Adaptive Sampling”). An ablation for iterative semantic refinement, which is applied during  the sequential optimization stage, has also been presented in Table 3 (“Full Initialization”). In addition, we show the ablation results without confidence map here, which achieves 89.4 $\mathcal{J}$ & $\mathcal{F}$ and 48.3 $<\delta_{avg}$. This shows that our confidence map is effective in improving the performance. We will make the contribution of each component more clearly in the final paper.
>
>
> **Q5: How does the adaptive resampling strategy handle regions that are under-represented due to occlusion?**
>
> **A5:** Our adaptive resampling strategy operates only on visible regions. Fully occluded areas cannot be directly reconstructed since their true geometry is inherently unobservable from the available data. Nevertheless, as shown by the tracking results in Figures 1 and 5, our 3D representation remains more stable in under-represented regions compared to the 2D networks. This suggests that our 3D representation is capable of producing plausible and coherent results by enforcing spatial and temporal consistency, even when direct observations are unavailable and 2D predictions are drifting.
>
> **Q6: Sequential-global optimization is resource-heavy and time-consuming.**
>
> **A6:** We acknowledge that achieving real-time performance during optimization is beyond our current scope. However, our primary goal is to produce accurate and temporally consistent results over long sequences, which is challenging for lightweight pre-trained 2D models. Despite the computational overhead, our runtime is comparable to other 3D-based pipelines: optimizing a scene with 100 frames typically takes approximately 30 minutes on a single 4090 GPU, close to the baseline of 4DGS (~24min). After optimization, the rendering is highly efficient, at about 75 fps, which enables real-time rendering and querying for segmentation and tracking results.

---

### Official Review · Reviewer_XuWq · 2025-06-26

**Clarity:** 3
**Significance:** 3
**Originality:** 3
**Rating:** 4
**Confidence:** 3

**Summary:**

This paper proposes Motion4D, a motion and semantics understanding framework. It lifts priors from 2D foundation models to 4D space with dynamic Gaussians. To alleviate the spatial misalignment and temporal flickering issues in 2D models, Motion4D proposes a set of technologies including iterative gt refinement, dual phases optimization and error-induced Gaussian densitification. To validate the effectiveness of the proposed method, this paper further introduces the DyCheck-VOS dataset. Extensive experiments are conducted.

**Questions:**

1. The iterative motion refinement module: the confidence $\omega(p)$ is learned adaptively, but its gt is manually designed. Can we replace it with a hand-crafted value? Like $\delta - ||I_t(p), I_{t'}(p) ||$
2. The iterative pipeline shows similar mechanism with SA3D [1]. More discussion about it is encouraged.

[1] Cen, J., Zhou, Z., Fang, J., Shen, W., Xie, L., Jiang, D., ... & Tian, Q. (2023). Segment anything in 3d with nerfs. Advances in Neural Information Processing Systems, 36, 25971-25990.

**Ethical Concerns:**

["NO or VERY MINOR ethics concerns only"]

**Final Justification:**

The response addresses most of my concern thus I keep the score.

**Limitations:**

Yes.

**Quality:**

3

**Strengths And Weaknesses:**

Strengths:

1. The method seems solid. Experimental results are good.
2. A new dataset is provided, providing good for following studies.
3. Writing is clear.

Weaknesses:

1. The method is complex, making it hard for reproducing.
2. Also, the complexity of the method makes me doubt about the robustness of it. Though the ablation study demonstrate the effectiveness of each component independently, it can not provide any evidence for the generalization ability of the proposed method. Can the method be simplified?
3. Figure 2 is important but unclear. The figure looks good but cost me for some time to totally understand. Some more annotations can be added to enhance its readability.
4. The proposed method requires per-scene optimization, which brings time overhead compared with e2e methods. It is essential to analyze its time consumption.
5. Table 3 is unclear. I think there is a typo. Check the check mark of the row of w/o global optimization.

---

> ### Author Rebuttal · Authors · 2025-07-31
>
> **Responses to Reviewer XuWq**
>
> **Q1: The method is complex and hard to reproduce.**
>
> **A1:** Our design is intentionally modular to ensure simplicity and ease of replication. Each component (SAM2, Track Any Point, Depth Anything, and 4DGS) is an independent and standalone module, which makes the overall pipeline straightforward to reproduce. Our primary contribution is the unified optimization framework that connects these modules. This framework is conceptually simple, effective, and clearly detailed in Sections 3.2–3.4. We have also indicated in Line 20 of the paper that we will release the complete codebase including training and evaluation scripts upon acceptance to ensure full reproducibility.
>
> **Q2: Robustness and generalization of the method.**
>
> **A2:** The modular design of our method enhances robustness. Large pre-trained networks such as SAM2 and Track Any Point are known for their strong generalization across diverse datasets. By using these models, we improve the robustness of the 4DGS-based scene representation. As shown in Tables 1 and 2, previous 3D representation methods often fail to generalize to casual dynamic scenes in the DAVIS and DyCheck datasets, where uncontrolled motion and noise violate their assumptions. Although 2D networks generalize well in many scenarios, Figures 1 and 4 demonstrate their limitations under severe occlusions and complex dynamics. In such cases, 3D reasoning is essential to achieve consistent and reliable results.
>
>
> **Q3: Figure 2 is unclear.**
>
> **A3:** We appreciate the reviewer’s feedback regarding Figure 2. In the revised version, we will improve its readability by adding annotations to key components and refining the caption to better explain the pipeline.
>
> **Q4: Time consumption analysis.**
>
> **A4:** Our method differs from single-task end-to-end 2D models by building a full 4D representation that jointly produces segmentation masks, tracking, depth and rendered images for both training and novel views. This richer capability inherently requires per-scene optimization, as we model the scene’s 3D geometry and dynamics jointly rather than processing frames independently. Despite the joint optimization, our runtime remains comparable to other 3D-based pipelines. Optimizing a scene with 100 frames typically takes about 30 mins on a single 4090 GPU, which is close to the 4DGS baseline (~24mins). After optimization, the rendering is highly efficient, at about 75 fps, which enables real-time rendering and querying for segmentation and tracking results. We will clarify by adding a detailed runtime analysis in our final paper.
>
> **Q5: Table 3 is unclear.**
>
> **A5:** We thank the reviewer for pointing this out. There is indeed a typo in Table 3, which we will correct.
>
> **Q6: Replacing the confidence $w(p)$ with a hand-crafted value.**
>
> **A6:** The confidence $w(p)$ is an adaptively learned logit, with its ground-truth defined based on the rendered result. This is similar to TAP model designs, where the predicted confidence logit is supervised by a binary signal, which is set to 1 when the prediction is within a predefined threshold and 0 otherwise. This binary signal serves only as a training target and enables the model to learn a smooth confidence field in 3D space. In contrast, hand-crafted confidence measures often yield brittle and noisy weights, particularly in regions with similar textural structures.
>
> **Q7: Comparison with SA3D.**
>
> **A7:** Our iterative pipeline shares a similar prompting mechanism with SA3D, but differs substantially in design and scope:
>
> - **(a) Target setting:** While SA3D is designed for static scenes and focuses solely on propagating a single-view mask to multi-view masks, our method targets monocular videos in dynamic scenes and refines SAM masks across the entire sequence to construct a coherent semantic representation.
>
> - **(b) Temporal consistency:** In addition to multi-view consistency, our method explicitly enforces temporal consistency by combining SAM prompting with sequential optimization of semantics as shown in Figure 3(b). This enables masks to propagate smoothly across frames and align segmentation with dynamic motion.
>
> - **(c) Iterative refinement:** Our approach jointly refines both the 2D priors and the 3D semantic field, instead of only updating a semantic voxel grid as in SA3D. This design allows us to progressively improve mask quality across frames and views by re-prompting SAM using the corrected inputs. In comparison, SA3D queries SAM only once per frame with no feedback loop.
>
> We will include this discussion and compare our method with SA3D in the final paper.

---

> > ### Comment · Reviewer_XuWq · 2025-08-05
> >
> > Thanks for the detailed response. My concerns are well addressed and thus I hope to maintain the borderline accept.

---

### Official Review · Reviewer_D7AY · 2025-06-28

**Clarity:** 4
**Significance:** 3
**Originality:** 3
**Rating:** 4
**Confidence:** 4

**Summary:**

In this paper, the authors propose an iterative optimization framework to jointly refine the 4D motion prediction and segmentation results of videos. They directly start with 2D tracks, depth prediction, and segmentation maps from well-established pre-trained models. Then they propose to construct the scene with 3DGS representation, which is enriched with semantic features and motion deformation field. The paper then proposes an iterative motion refinement technique with an adaptive resampling strategy and an interactive iterative semantic refinement module. For optimization, they develop sequential optimization and global optimization to build both short and long temporal relations.

**Questions:**

1. It would be better if the authors could provide visualization comparisons of ablation settings (just like column (d)(e) in Figure 4 clearly showing the improvement from rendered masks to refined SAM2 masks). For example, the authors mentioned that without global optimization, there is temporal drift across chunks. However, this is not evident from mere numerical results. A visualization comparison would be more intuitive.

2. I am curious about why not optimizing depth estimation. Depth is a crucial part of 3DGS scene reconstruction and point tracking.

3. Since point track and depths are both available, it would be better if the authors could compare their results with 3D point tracking benchmarks.

4. I'm not very familiar with video segmentation, but I'm curious about why not evaluate the mIoU metric for segmentation results.

5. What is the computational cost (time, gpu, params) of each optimization process? Since this is a refinment method, high fps and lightweight are crucial.

**Ethical Concerns:**

["NO or VERY MINOR ethics concerns only"]

**Final Justification:**

The authors have addressed most of my concerns, and as a result, I maintain my original positive score. I hope the authors could incorporate the points discussed in the rebuttal into the revised version of their paper.

**Limitations:**

Yes.

**Paper Formatting Concerns:**

N/A.

**Quality:**

3

**Strengths And Weaknesses:**

#### Strengths:

1. The paper is overall well-written and easy to follow. The motivation is clearly stated.

2. The idea of jointly optimizing 4D motion and semantic segmentation via 3DGS representation is interesting, novel, and technically sound.

3. The quantitative comparisons with the previous method and the visualization results strongly convince the effectiveness of the proposed method.

#### Weaknesses:

1. The overall concept of the paper is rather incremental. Currently, the community is expecting systematic design with a unified framework and joint optimization of multiple tasks. I acknowledge the efforts made by the authors to jointly optimize low-level 3D motion prediction and high-level semantic segmentation. Their technique is also effective. However, the paper still seems like an auxiliary refinement module to be inserted in pre-trained models. I like the idea of unifying with 3DGS representation, but it would be better if the authors could draw a bigger picture with this concept, rather than just implement it as a refinement module.

2. The experiment part is not thorough enough. It would be better if more visualization comparisons about each ablation setting could be provided to vividly show their effectiveness.

---

> ### Author Rebuttal · Authors · 2025-07-31
>
> **Responses to Reviewer D7AY**
>
> **Q1: The overall concept of the paper is rather incremental.**
>
> **A1:** We thank the reviewer for the valuable feedback. While Motion4D leverages existing 2D and 3D components, its contribution goes beyond serving as an auxiliary refinement module. Motion4D constructs a joint 4D representation that integrates motion, depth, and semantic cues. This directly addresses the reviewer’s call for “a unified framework and joint optimization of multiple tasks”. Crucially, the joint optimization of 2D priors and an explicit 3D representation is not only complementary but fundamental to the effectiveness of our approach. We will revise the final paper to clarify and emphasize that Motion4D is a carefully designed joint optimization framework rather than a simple refinement pipeline.
>
> **Q2: Visualization of ablation settings.**
>
> **A2:** We appreciate the suggestion to include additional visualizations and will incorporate them in the final paper. Although figures cannot be attached in this rebuttal, we highlight the key findings from our ablation studies. Particularly, removal of the adaptive resampling module leads to unreliable reconstruction in fast-moving or occluded regions such as the legs of dynamic objects. Figure 3 of the supplementary material shows the importance of this component, where the baseline without resampling exhibits noticeable artifacts and even missing regions in the challenging regions. In addition, without global optimization, we observed that the Gaussians may be shifted abruptly between chunks, leading to clear motion drift.
>
> **Q3: Optimizing depth estimation.**
>
> **A3:** Depth is optimized jointly with 2d tracks as part of the iterative motion refinement process. Specifically, Equation (5) defines the depth loss as the discrepancy between the 3D-rendered depth and the 2D depth priors, formulated in a manner consistent with the loss for 2D tracks.
>
> **Q4: Comparison on 3D point tracking benchmarks.**
>
> **A4**: We have evaluated 3D point tracking performance on the DyCheck benchmark, which includes tracking annotations and LiDAR depth. The results are reported in Table 1 of the supplementary material. Our Motion4D demonstrates its ability to maintain accurate long-term 3D correspondences by achieving consistent improvements over existing 3D methods. We will emphasize these results and provide more detailed discussion in the final paper.
>
> **Q5: Evaluating the mIoU metric for segmentation results.**
>
> **A5:** Our evaluation follows the standard $\mathcal{J}$ & $\mathcal{F}$ metric widely adopted in video object segmentation. Here, the $\mathcal{J}$ score corresponds to the mean Intersection over Union (mIoU) across frames, and the $\mathcal{F}$ score measures contour accuracy. This means that our reported $\mathcal{J}$ & $\mathcal{F}$ results have included mIoU which ensures comparability with other VOS benchmarks.
>
> **Q6: Computational cost of the optimization process.**
>
> **A6:** Our model is trained and evaluated entirely on a single RTX 4090 GPU and contains approximately 150M parameters, depending on the scope of the scene. The training phase of our Motion4D is not real-time since it relies on predictions from multiple pre-trained networks and follows the per-scene optimization paradigm of 3DGS-based methods. Despite this, our runtime is comparable to other 3D-based pipelines: optimizing a scene with 100 frames typically takes approximately 30 minutes on a single 4090 GPU, close to the baseline of 4DGS (~24min). The rendering of segmentation and tracking results is fast, at about 75 fps, after optimization since the 4DGS representation can be rendered efficiently. We will clarify in the final paper that real-time performance during optimization is beyond our current scope. The primary goal of our paper is to deliver accurate and temporally consistent results over long sequences, which is challenging for lightweight pre-trained 2D models.

---

> > ### Comment · Reviewer_D7AY · 2025-08-05
> >
> > Thanks the authors for their response. Their reply has addressed most of my concerns. Regarding Q4 on 3D point tracking benchmarks, it would be preferable for the authors to include experiments on the more widely adopted TAP-Vid3D benchmark, and to compare against methods such as DELTA, SpatialTracker, etc. However, given the limited discussion period, this is not mandatory. Or a small-scale experiment on a representative subset (e.g., DriveTrack, Aria, or PStudio) would also be acceptable.

---

> ### Author Response · Authors · 2025-08-08
>
> We appreciate the reviewer's suggestion to add new experiments on the TAPVid-3D benchmark, and we will include them in the final paper. However, due to limited time during the discussion period, we were only able to provide a experiment on the "minival" split of the DriveTrack and PStudio subsets:
>
> |              | $\|$ |           | **DriveTrack** |           |$\|$ |           | **PStudio** |           |
> |:---------|:-:|:---------:|:--------------:|:---------:|:-:|:---------:|:-----------:|:---------:|
> |Methods | $\|$ | 3D-AJ $\uparrow$ | APD $\uparrow$     | OA $\uparrow$     |$\|$ | 3D-AJ $\uparrow$  | APD $\uparrow$      | OA $\uparrow$     |
> |SpatialTracker | $\|$ |  5.8       | 10.2           | 82.0      | $\|$ | 9.8       | 17.7        | 78.4     |
> |Ours | $\|$ |  10.3       | 16.0           | 83.1      | $\|$ | 10.4       | 18.1        | 82.0     |
>
> Here, we report the 3D Average Jaccard (3D-AJ), the average percent of points within $\delta$ error (APD), and the occlusion accuracy (OA). As shown in the table, our method outperforms SpatialTracker across all metrics for 3D tracking performance. We believe this improvement stems from our explicit 3D representation, which allows the method to leverage spatial geometry more effectively than pure 2D networks.

---

> > ### Comment · Reviewer_D7AY · 2025-08-09
> >
> > Thank you for providing such detailed answers to my questions. I will take them into consideration in forming my final evaluation.

---

### Official Review · Reviewer_iUVL · 2025-06-30

**Clarity:** 2
**Significance:** 4
**Originality:** 4
**Rating:** 5
**Confidence:** 3

**Summary:**

This submission presents a method for simultaneously reconstructing and tracking semantics and points in a 4D scene given input posed RGB imagery. The method leverages pretrained SAM2, Track Any Point, and Depth Anything models for segmentation, depth, and tracking priors. A 3DGS model with temporally varying gaussians is optimized via a series of optimization stages (first sequential motion refinement, then sequential semantic refinement, then a global optimization where all frames and layers are optimized together. The method demonstrates impressive results.

**Questions:**

1. Please further explain the relationship between the confidence and uncertainty maps. Please visualize if possible.

2. Please further explain the global motion bases.

3. Please add visualizations if possible to help clarify how the optimization progresses over time (from a novel view).

**Ethical Concerns:**

["NO or VERY MINOR ethics concerns only"]

**Final Justification:**

All other reviews concur with initial accept recommendation. Author rebuttal is reasonable and no other major concerns were raised during rebuttal process.

**Limitations:**

The limitations section could definitely be more extensive. What are some failure cases? Do any of those occur in the DyCheck-VOS dataset?

**Quality:**

3

**Strengths And Weaknesses:**

This is a compelling submission with strong results (on a modest number of scenes) and relatively thorough ablations. The proposed system is complex but manages to simultaneously optimize multiple per-frame priors into a coherent 4D whole in a way that I have not seen before. I don't personally know of anything immediately comparable, though other reviewers with more experience in this area may.

I do have some clarity and experimental questions/concerns, but I do not think they are severe enough to warrant rejection.

First, I do not fully understanding the confidence and uncertainty maps nor their exact relationship. I think it would be very helpful to see some video results that show the optimization stages progressing along with the underlying 4DGS representation shown from multiple views and colored by different fields. Without that, even a good text description makes it hard to really grasp the method.

Additionally, the deformation field is a little confusing as well-- the notation doesn't really make it easy to understand the relationship between # of bases, # of gaussians, and # of timestamps. There is also no ablation regarding this step, nor an easy intuition for why it is required.

Third, the proposed evaluation is quite small-- the DyCheck-VOS dataset is only 14 sequences. Only a handful of stills from those are shown, making it hard to understand the breadth of the data and how challenging it is (or isn't).

---

> ### Author Rebuttal · Authors · 2025-07-31
>
> **Responses to Reviewer iUVL**
>
> **Q1: Lack of visualization of confidence and uncertainty maps.**
>
> **A1**: We will include these visualizations in the revised version, but cannot provide them here due to the rebuttal’s format constraints. The "uncertainty logits" refer to the learnable attributes attached to the 3D Gaussians (one uncertainty logit per 3D Gaussian). We use this 3D field (the learnable attributes) to render 2D views of the confidence maps, which is similar to how 3D Gaussian colors are rendered into images. This approach allows us to optimize a 3D-consistent uncertainty field that reflects the underlying scene geometry, and use the rendered confidence maps to update 2D priors.
>
> **Q2: Lack of visualization of the optimization progress.**
>
> **A2**: We will include this visualization in the final paper. Nonetheless, we note that Figure 3 has provided a related illustration of the optimization progress. The figure shows our Motion4D progressively refines the inconsistent 2D priors and produces temporally consistent results during the sequential optimization stage.
>
> **Q3: The deformation field is a bit confusing.**
>
> **A3**: The deformation field follows the formulation introduced in prior work [1]. Instead of defining $N$ bases, where $N$ is the number of Gaussians, we define $B$ global bases ($B \ll N$), where each base contains $T$ elements in $\mathbb{SE}(3)$. We assign a coefficient vector of length $B$ for each 3D Gaussian, whose per-frame transformation is then computed as a weighted sum of the $B$ elements in $\mathbb{SE}(3)$ at that time step. This design significantly reduces the number of parameters and enables smooth temporal deformation. Although we adopt the same deformation field formulation, we extend it to integrate with the adaptive resampling module in Motion4D. This is why we kept its description concise in the method section. We also evaluated a variant using per-Gaussian transformation. While this increases flexibility, it results in slightly worse performance ($<\delta_{avg}$ drops from 50.4 to 49.6 on DyCheck) due to overfitting and unstable tracking estimation. In the final paper, we will expand the explanation and add this ablation to clarify its effect.
>
>
> **Q4: The proposed evaluation is quite small.**
>
> **A4**: The proposed DyCheck-VOS benchmark consists of long video sequences, each with 200–500 frames. For comparison, the DAVIS 2017 validation set for segmentation and tracking includes 30 short videos, each with fewer than 100 frames. As a result, DyCheck-VOS introduces greater challenges in long-term tracking, especially in handling frequent occlusions, spatial alignment, and object re-identification over extended sequences. Due to the difficulty of annotating dense point trajectories for Track-Any-Point, a larger benchmark is currently unavailable for evaluation. We will include additional examples to better showcase the complexity and challenges posed by Dycheck-VOS.
>
> **Q5: Failure cases on DyCheck-VOS.**
>
> **A5**: Our method may fail in scenarios where the underlying 3D reconstruction is unreliable, typically due to inaccurate SfM camera poses or extremely challenging scene geometry. Nonetheless, our proposed iterative refinement framework can address several failure cases when the 2D priors are noisy and unreliable. These cases are shown in Figure 3, where the input segmentation and tracking results are poorly estimated. In the final paper, we will elaborate on the causes of such failure cases and discuss potential strategies for mitigating them in future work.
>
> [1] Qianqian Wang, Vickie Ye, Hang Gao, Jake Austin, Zhengqi Li, and Angjoo Kanazawa. Shape of motion: 4d reconstruction from a single video. arXiv preprint arXiv:2407.13764, 2024.

---

### Decision · Program_Chairs · 2025-09-17

**Decision:**

Accept (poster)

**Comment:**

This paper tackles the problem of dynamic 3D reconstruction and proposed a framework which integrates 2D priors into a 4D representation in the form of a dynamic 3dgs. This final representation is optimized through a two stage process which jointly refines all attributes for extended time frame. This representation is then tested on several different downstream tasks with promising results.

The reviewers were positive (3 borderline accept, 1 accept) with the reviewers commending the overall framework as well as results. Reviewers were concerned about the technical novelty of the paper as well clean ablations on the component parts of the framework. However, all were positive in the end.

I also advocate for acceptance.